# A 0.15-to-0.5 V Body-Driven Dynamic Comparator with Rail-to-Rail ICMR

**Riccardo Della Sala** [ID], **Valerio Spinogatti** [ID], **Cristian Bocciarelli** [ID], **Francesco Centurelli** *[ID] and **Alessandro Trifiletti** [ID]

Dipartimento di Ingegneria dell'Informazione, Elettronica e Telecomunicazioni (DIET), Università di Roma La Sapienza, 00184 Roma, Italy; riccardo.dellasala@uniroma1.it (R.D.S.); valerio.spinogatti@uniroma1.it (V.S.); cristian.bocciarelli@uniroma1.it (C.B.); alessandro.trifiletti@uniroma1.it (A.T.)

* francesco.centurelli@uniroma1.it

**Abstract:** In this paper, a novel dynamic body-driven ultra-low voltage (ULV) comparator is presented. The proposed topology takes advantage of the back-gate configuration by driving the input transistors' gates with a clocked positive feedback loop made of two AND gates. This allows for the removal of the clocked tail generator, which decreases the number of stacked transistors and improves performance at low $V_{DD}$. Furthermore, the clocked feedback loop causes the comparator to behave as a full CMOS latch during the regeneration phase, which means no static power consumption occurs after the outputs have settled. Thanks to body driving, the proposed comparator also achieves rail-to-rail input common mode range (ICMR), which is a critical feature for circuits that operate at low and ultra-low voltage headrooms. The comparator was designed and optimized in a 130-nm technology from STMicroelectronics at $V_{DD} = 0.3$ V and is able to operate at up to 2 MHz with an input differential voltage of 1 mV. The simulations show that the comparator remains fully operational even when the supply voltage is scaled down to 0.15 V, in which case the circuit exhibits a maximum operating frequency of 80 kHz at $V_{id} = 1$ mV.

**Keywords:** comparators; body-driven; ultra-low voltage; ultra-low power; IoT

## 1. Introduction

Recent years have seen an increasing diffusion of electronic apparatuses such as laptops, tablets and smartphones to ease plenty of tasks, such as banking, booking, traveling, smart-working and so on. These portable systems are battery-powered or harvest their energy from the environment; thus, it is essential to optimize the power consumption by reducing both current levels and supply voltage [1–4]. One of the key requirements that devices must ensure is their autonomy, or the amount of time they can operate without requiring a recharge. This is an important characteristic that is often considered by users when selecting a device, and it has significant implications for the usability and convenience of the device in various settings. The ability of a device to provide sustained and reliable performance over an extended period of time is critical for its overall functionality and user satisfaction.

Moreover, in the biomedical field, there is growing interest in smart devices that can ease the study and diagnosis of neural disorders [5–11]. Recently, research has focused on implantable electronic apparatuses capable of monitoring and diagnosing neural disorders, such as Parkinson's, Alzheimer's, epilepsy and so on [12,13]. Typically, these complex systems are implanted under the scalp, on the neural tissue, and it is essential to minimize power dissipation, given the fact that an overheating of the system could irreversibly damage the neural tissue of the patient [14,15]. Given these considerations, the biomedical signal acquisition system should be able to operate at supply voltages as low as 0.3 V, with current consumption in the order of nA to guarantee very good autonomy and low temperatures.

Analog-to-digital converters (ADCs) are among the most important blocks in biomedical signal acquisition systems and, more generally, in smart devices that often process in the digital domain data acquired by analog sensors. Several ADC architectures have been proposed in the literature; in the biomedical and IoT contexts, where low-frequency signals must be digitized with medium-high resolution and extremely low power consumption, the most commonly used are the successive approximation register (SAR) ADC [16–29] and the sigma-delta ADC [30–34]. A key element of all these architectures is the comparator, whose purpose is to determine the sign of the input differential voltage, providing an output that can be interpreted as a logic 0 or 1 level. The different ADC architectures pose different requirements on the comparator, whose main performances are related to speed, resolution and power consumption. Dynamic clocked comparators are typically used to allow synchronization and minimize power consumption.

The speed of the comparator is determined by the delay from the clock edge to a stable logic output level and is related to the ADC sampling speed taking into account the ADC architecture. The comparator resolution is determined both by its sensitivity, i.e., the minimum voltage difference that can be detected, and by the offset. A further important feature in many applications is the input common-mode range (ICMR): some ADC architectures (e.g., exploiting single-ended signals [35,36], or SAR ADCs based on the set-and-down algorithm [37]) require a rail-to-rail ICMR to allow a comparison of signal levels across the whole input range [36,38–40].

Several latched comparators have been proposed in the literature that can operate with supply voltages of 0.5 V and lower [21,24,30,31,41,42]. They are often based on the StrongARM architecture [17–20,25–27,38,43,44], where a differential pair with a clocked tail current generator is loaded by a pair of cross-coupled inverters that form a latch. The ultra-low-voltage (ULV) operation is often achieved by substituting the inverters with simple PMOS devices [28,34,45–47], cross-coupled in positive feedback, or by exploiting body driving. In this case, the body terminals of the NMOS devices of the inverters can be used as input terminals, and this allows eliminating the tail current generator or the differential pair [16,22,23,45,46,48,49]. Akbari in [48] proposed a body-driven dynamic comparator where PMOS devices were used both as cross-coupled latch and as input devices, and NMOS transistors are exploited for dynamic biasing and to reset the output; the comparator is able to operate with a supply voltage as low as 0.3 V. Still lower supply voltages are reported by Yang et al. [44], that present a StrongARM comparator operating in subthreshold with an auxiliary amplifier, operating down to 0.25 V, and by Li [47], where a gate-driven StrongARM exploiting cross-coupled PMOS devices as the latch is simulated down to 0.2 V supply. A StrongARM latch exploiting both gate driving and body driving and a boosted clock has been reported by Zhou et al. in [16] in a SAR ADC operating at 0.16 V supply.

Recently, research in the field of fully-synthesizable analog circuits to ease mixed-signal integration and portability, and to minimize time-to-market, is gaining popularity [50–56]. Standard-cell-based comparators have been reported operating down to 0.15 V supply, and are, therefore, a suitable alternative to analog circuits for ULV operation. The simplest latched comparator can be implemented using cross-coupled 3-input NAND gates [57]. This configuration provides a very compact comparator but with a limited ICMR, therefore, architectures that pair both NAND and NOR gates as input cells [36,58–60], or are based on AND-OR-INVERTER (AOI) gates [61], have been proposed to achieve a rail-to-rail ICMR, at the cost of higher complexity and area footprint.

In this paper, we present a ULV StrongARM comparator topology that allows rail-to-rail ICMR through body-driving, very-low supply voltages by minimizing the number of stacked devices and good speed and power performance. A trade-off between offset and area footprint yields suitable offset performance with an optimized area that is comparable with standard-cell-based implementations. The proposed topology is described in Section 2 and analyzed in Section 3. Simulation results in a commercial 130-nm CMOS technology are

reported in Section 4 and compared with the state-of-the-art in Section 5. Finally, Section 6 concludes.

## 2. Proposed Comparator Topology

The proposed ULV StrongARM comparator, shown in Figure 1, exploits body driving to achieve rail-to-rail ICMR and minimizes the number of stacked devices to allow operation down to very low supply voltages. Moreover, limiting the number of stacked devices allows for maximizing the drain–source voltages of the transistors for a given supply voltage, resulting in a speed improvement. Inverters in the standard StrongARM topology are substituted by simple cross-coupled PMOS devices, and the clocked tail current generator is eliminated, achieving a core comparator with just two stacked devices.

Without the tail current, the gates of the NMOS input devices have to be used to turn on and off the input pair in the evaluation and reset phases, respectively. However, applying the clock signal directly to the gates of the input pair results in excessive power dissipation. With reference to Figure 1, but with the clock signal directly applied to the gates of $M_{1,2}$, we have that in the evaluation phase the comparator behaves as a body-driven pseudo-differential pair with cross-coupled PMOS devices in positive feedback. The input signal modulates the drain current of $M_{1,2}$ so that output voltages $V_p$ and $V_q$ start decreasing from the initial value $V_{DD}$ at different rates until one of the load devices, $M_{3,4}$, turns on. At this point, a DC path between $V_{DD}$ and the ground is established and remains active for the rest of the evaluation phase.

To solve this problem, the solution presented in Figure 1 is proposed: the gates of the input devices $M_{1,2}$ are driven by cross-coupled AND gates that perform a logic-AND operation between the outputs and the clock signal. When the clock signal is low (voltage equal to 0), the outputs of the AND gates set the gates of $M_{1,2}$ to ground, turning off the input pair. At the beginning of the evaluation phase, the clock signal increases (voltage equal to $V_{DD}$), and the outputs $V_{p,q}$ have been precharged to $V_{DD}$; thus, the AND gates yield a logic 1, i.e., they set the voltage on the gates of $M_{1,2}$ to $V_{DD}$. Once one of the outputs (e.g., $V_p$) drops enough to be interpreted as logic 0, the corresponding AND gate yields a logic 0 that turns off $M_2$, thus, interrupting the dc current path to ground (current in the other path is zero, since $M_3$ is off). The comparator core must be followed by an SR-latch to keep the outputs during the reset phase.

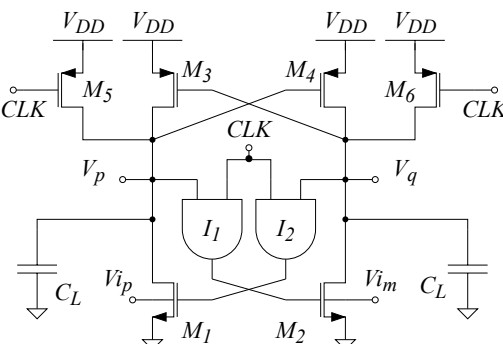

**Figure 1.** Proposed ULV comparator.

The operation of the proposed comparator can be synthesized as follows:

- *Reset Phase*: in the reset phase, the clock is low; hence the two NMOS $M_{1,2}$ are turned off, whereas the two PMOS $M_{5,6}$ pump current in the output nodes $V_{p,q}$, charging the output parasitic capacitances to $V_{DD}$. In this phase, the PMOS-driven SR-latch holds the previous data due to the positive feedback of $M_{9,10}$.
- *Evaluation Phase*: in the evaluation phase, the two NMOS are turned on, whereas the PMOS $M_{5,6}$ are turned off. In this phase, given the difference between the two body voltages of $M_{1,2}$, a differential current is generated, and the positive feedback forced by $M_{3,4}$ sets the outputs to $V_{DD}$ and ground according to the sign of the input

differential signal. In this condition, the feedback through the AND gates turns off one of the input devices, $M_{1,2}$, thus, avoiding static power consumption. The SR-latch senses the differential output voltage $V_p - V_q$ and unbalances the outputs accordingly.

## 3. Analysis of the Delay

In this section, the operation of the proposed comparator is examined from a theoretical standpoint, and an analytical estimation of the delay is derived. The analysis can be carried out by observing that the evaluation phase can be divided into two sub-phases: preamplification and regeneration. Preamplification refers to the time interval from the rising edge of the clock signal to the instant when $M_{3,4}$ turns on. Regeneration corresponds to the time interval from the end of preamplification to when the signal is regenerated at the outputs.

### 3.1. Preamplification

During preamplification, the comparator is described by the equivalent model shown in Figure 2.

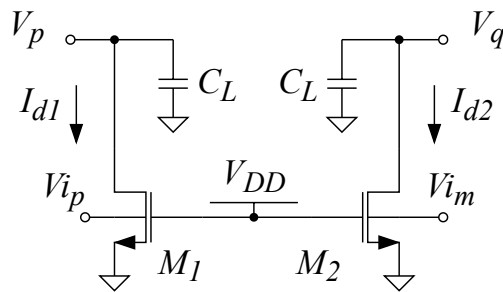

**Figure 2.** Equivalent circuit of the proposed comparator during preamplification.

Preamplification lasts until the output common mode voltage $V_{ocm} = (V_p + V_q)/2$ drops to a level that allows the PMOS load to turn sufficiently on. Since devices are operating in subthreshold, the source–gate voltage to be considered cannot be the MOS threshold voltage. Hence, we arbitrarily define the voltage $V_{LIM}$ as the voltage at which the drain current of $M_{3,4}$ equals that of $M_{1,2}$. Assuming a small differential input voltage $V_{id} = V_{i_p} - V_{i_m}$, we can approximately consider that the preamplification phase ends when the output common mode voltage $V_{ocm}$ equals $V_{DD} - V_{LIM}$. Therefore, the length of this phase is given by

$$t_{pre} = \frac{C_L V_{LIM}}{I_{cm}} \tag{1}$$

where $I_{cm}$ is the common mode drain current, defined as $(I_{d1} + I_{d2})/2$.

During preamplification, $M_1$ and $M_2$ discharge asymmetrically nodes P and Q, thus, integrating the input difference $V_{id}$ on the output nodes. The output differential voltage at the end of preamplification, which we denote as $V_{od}^{pre}$, can be expressed as

$$V_{od}^{pre} = V_p^{pre} - V_q^{pre} = \frac{Q_{od}}{C_L} = \frac{I_{dm} t_{pre}}{C_L} = \frac{I_{dm} V_{LIM}}{I_{cm}} = \frac{g_{mb} V_{LIM}}{I_{cm}} V_{id} \tag{2}$$

where $Q_{od}$ is the difference between the amounts of the charge stored in the parasitic capacitances at nodes P and Q, and $I_{dm}$ is the differential drain current, defined as $I_{d1} - I_{d2}$. The preamplification gain is given by

$$A_v^{pre} \triangleq \frac{V_{od}^{pre}}{V_{id}} = \frac{g_{mb} V_{LIM}}{I_{cm}} \tag{3}$$

where $g_{mb}$ is the body transconductance. It is worth noticing that, differently from the standard StrongARM comparator, $I_{d1}$ and $I_{d2}$ can be considered constant without approximations because the devices' $V_{gs}$ and $V_{bs}$ are constant.

### 3.2. Regeneration

During regeneration, the positive feedback loop formed by $M_{3,4}$ starts to regenerate the signal. The AND gates, which have been represented in figure 3 as buffers because CK = $V_{DD}$, help speed up regeneration because they turn off either $M_1$ or $M_2$, depending on the sign of the input difference. If the inverting amplifiers formed by $M_{1,3}$ and $M_{2,4}$ operate in the linear region of their transcharacteristic, the behavior of the comparator can be described in the Laplace domain by the linearized circuit shown in Figure 4. The initial conditions are accounted for by the generators $C_L V_p^{pre}$, $C_L V_q^{pre}$, $CV_{g1}^0$, $CV_{g2}^0$ and $V_{DD}/s$ (the power supply). For the sake of simplicity, the buffers are modeled as first-order systems that consist of $g_{mA}$ and $C$. This is obviously an approximation because the AND gates should be represented as the cascade of two inverters. Moreover, the output resistances of the devices have been neglected. Note that the $g_m$ of the PMOS devices is assumed to be identical to the $g_m$ of the NMOS devices because the aspect ratios of $M_{1,2}$ and $M_{3,4}$ are chosen in such a way that $g_{m,p} = g_{m,n}$ at the beginning of the regeneration phase.

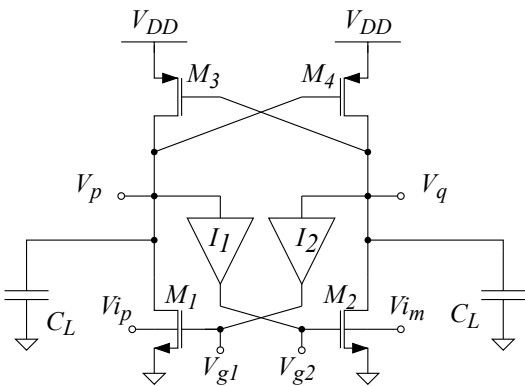

**Figure 3.** Equivalent circuit of the proposed comparator during regeneration.

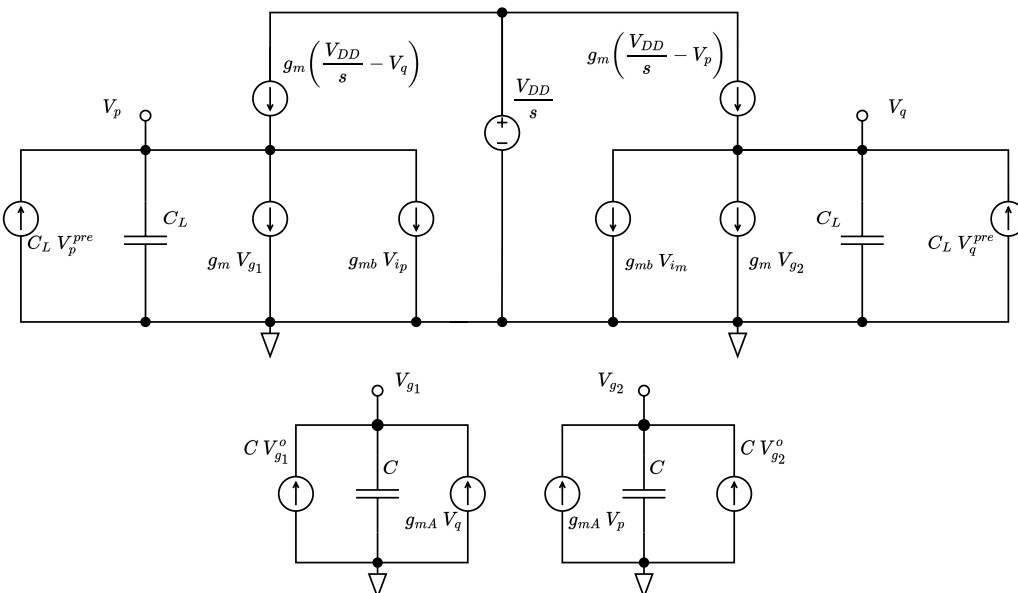

**Figure 4.** Equivalent circuit of the proposed comparator during regeneration, with explicit transistor models.

By analyzing the circuit in Figure 4, the following equations are obtained:

$$g_m(V_{DD}/s - V_q) = sC_L V_p - C_L V_p^{pre} + g_m V_{g1} \tag{4}$$

$$g_m(V_{DD}/s - V_p) = sC_L V_q - C_L V_q^{pre} + g_m V_{g2} \tag{5}$$

$$g_{mA} V_q = sCV_{g1} - CV_{g1}^0 \tag{6}$$

$$g_{mA} V_p = sCV_{g2} - CV_{g2}^0 \tag{7}$$

The currents $g_{mb}V_{i_p}$ and $g_{mb}V_{i_m}$ have been neglected because they cause $V_{od}$ to increase at a constant rate, which means their role is secondary with respect to the exponential behavior that results from positive feedback. Moreover, $g_{mb}$ is several times smaller than $g_m$. By subtracting Equation (5) from Equation (4) and Equation (7) from Equation (6) one obtains the system

$$\begin{cases} g_m V_{od}(s) = sC_L V_{od}(s) - sC_L V_{od}^{pre} + g_m V_{gd}(s) \\ -g_{mA} V_{od}(s) = sCV_{od}(s) - sCV_{gd}^0 \end{cases} \tag{8}$$

where $V_{od} \triangleq V_p - V_q$ is the output differential voltage and $V_{gd} \triangleq V_{g1} - V_{g2}$ is the differential signal at the gate terminals of $M_{1,2}$. It should be noted that at the end of preamplification, we have $V_{g1} = V_{g2} = V_{DD}$, which implies $V_{gd}^0 = 0$ V.

By solving (8) for $V_{od}(s)$ one has

$$V_{od}(s) = \frac{sV_{od}^{pre}}{s^2 - s\frac{g_m}{C_L} - \frac{g_m g_{mA}}{CC_L}} \tag{9}$$

The poles of $V_{od}(s)$ are given by

$$p_{1,2} = \frac{g_m}{2C_L} \left( 1 \pm \sqrt{1 + \frac{4\tau_L}{\tau_{AND}}} \right) \tag{10}$$

where $\tau_L \triangleq C_L/g_m$ and $\tau_{AND} \triangleq C/g_{mA}$. By letting

$$\delta \triangleq \sqrt{1 + \frac{4\tau_L}{\tau_{AND}}} - 1 > 0 \tag{11}$$

it follows that the two poles can be rewritten as

$$p_1 = \frac{g_m}{C_L}(1 + \delta) \tag{12}$$

$$p_2 = -\frac{g_m \delta}{2C_L} \tag{13}$$

By taking the inverse Laplace transform of (9) and manipulating the resulting expression, it is straightforward to show that

$$v_{od}(t) = \frac{V_{od}^{pre}}{2} \left( \frac{2+\delta}{1+\delta} e^{\frac{t}{\tau_1}} + \frac{\delta}{1+\delta} e^{-\frac{t}{\tau_2}} \right) \tag{14}$$

where $\tau_1 \triangleq 1/|p_1|$ and $\tau_2 \triangleq 1/|p_2|$. Two observations can be made:

- The first term on the left side of Equation (14) shows that the initial difference will be regenerated exponentially until the outputs are saturated. The second term, instead, vanishes as $t \to \infty$ and is, thus, negligible.
- It is easy to verify that if the AND gates were not used, the regeneration time constant would be $C_L / g_m$, which is greater than $C_L / (g_m(1 + \delta)) = \tau_1$. This confirms that not only the addition of the AND gate improves power consumption by cutting the conductive path between $V_{DD}$ and ground, but it also provides an advantage in terms of delay because it speeds up regeneration. The improvement with respect to the version without AND gates depends on the ratio $\tau_L / \tau_{AND}$: the smaller $\tau_{AND}$ is compared to $\tau_L$, the larger the reduction in delay that is obtained by adding the AND gates. This is in accordance with intuition because the advantage derived from the AND gates becomes larger as their delay decreases.

By neglecting the exponentially decreasing term in Equation (14), the regeneration time is easily obtained in closed form; its expression is

$$t_{reg} = \tau_1 \ln\left(\frac{V_{DD}(1 + \delta)}{V_{od}^{pre}(2 + \delta)}\right) = \frac{C_L}{g_m(1 + \delta)} \ln\left(\frac{V_{DD}(1 + \delta)}{V_{od}^{pre}(2 + \delta)}\right) \tag{15}$$

As usual, the regeneration time is defined as the amount of time required by the comparator to regenerate its output difference to $V_{DD}/2$, starting from the initial difference $V_{od}^{pre}$.

By combining (1) and (15), an analytical estimation of the comparator's total delay is obtained:

$$t_d = t_{pre} + t_{reg} = \frac{C_L V_{LIM}}{I_{cm}} + \frac{C_L}{g_m(1 + \delta)} \ln\left(\frac{V_{DD}(1 + \delta)}{V_{od}^{pre}(2 + \delta)}\right) \tag{16}$$

*3.3. Analysis of the Input Referred Offset vs. Mismatch Variations*

One of the main drawbacks of body-driven stages with respect to gate-driven ones is the input-referred offset, which is a main requirement in many comparators. In this section, it has been derived how the performance of body-driven stages is compromised by the low body transconductance gain.

In [62], a detailed analysis of the input-referred offset has been derived for gate-driven architecture as

$$\sigma^2(V_{GS}) \approx \sigma^2(V_T) = \frac{A_{V_T}^2}{W\,L} \tag{17}$$

where $\sigma(V_{GS})$ is the standard deviation of the gate–source voltage, $A_{V_T}$ is the area-proportional constant for the $V_{GS}$ voltage, and W and L are the width and the length of the transistor, according to Pelgrom's law [63]. Analogously one can derive the input-referred offset voltage for a body-driven architecture as

$$\sigma^2(V_{BS}) \approx \sigma^2(V_T) = \frac{A_{V_T}^2}{W\,L} \cdot \frac{1}{\alpha^2} \tag{18}$$

Here, a further term $\alpha$ was added to take into account the threshold voltage dependence with respect to $V_{BS}$ according to the following relation: $V_T = V_{th_0} - \alpha \cdot V_{BS}$. The $\alpha$ term is lower than 1; thus, it is clear that by considering equal transistor sizes, the body-driven stages would degrade their performance with respect to gate-driven ones for a given technology node or CMOS process.

## 4. Simulation Results

*4.1. Comparator*

The proposed comparator was designed in a commercial 130-nm technology by STMicroelectronics and simulated with Cadence Virtuoso. All simulations were carried out

by loading the outputs of the comparator with minimum area inverters. Transistor sizes have been chosen according to the trade-off between delay and input-referred offset. In detail, the aspect ratio and area of the NMOS devices were chosen to optimize delay while achieving an acceptable input-referred offset under mismatch variations. Then, the aspect ratio of the PMOS devices was chosen in such a way as to meet the condition $g_{m,n} = g_{m,p}$ when $V_{ocm} \approx V_{DD}/2$. This ensures that both outputs are fully regenerated (one to $V_{DD}$ and the other to ground), even for small input differences. Suppose, for instance, that $V_{id}$ is slightly larger than zero. If $g_{m,p}$ were too small compared to $g_{m,n}$, then $M_4$ would not be able to pull $V_q$ all the way up to $V_{DD}$. If, instead, $g_{m,p}$ were higher than $g_{m,n}$, then $M_1$ would not be able to pull $V_p$ down to the ground. The resulting device dimensions, optimized for operation at 0.3 V supply, are reported in Table 1. It is important to note that the use of body driving applied to NMOS devices requires a triple-well process and leads to an increase in overall area occupation. However, it should also be remarked that our design was optimized to minimize resource consumption as much as possible. The layout is reported in Figure 5 and shows an area of about 667.604 µm$^2$ (31.64 µm × 21.10 µm). Standard cells have been used for the AND gates.

**Table 1.** Transistor sizing of the proposed comparator.

| Device | W | L |
|--------|------|------|
|        | [µm] | [µm] |
| $M_{1,2}$ | 2 | 2 |
| $M_{3,4}$ | 40 | 0.2 |
| $M_{5,6}$ | 2 | 0.2 |
| $M_{7,8}$ | 5 | 1 |
| $M_{9,10}$ | 1 | 1 |

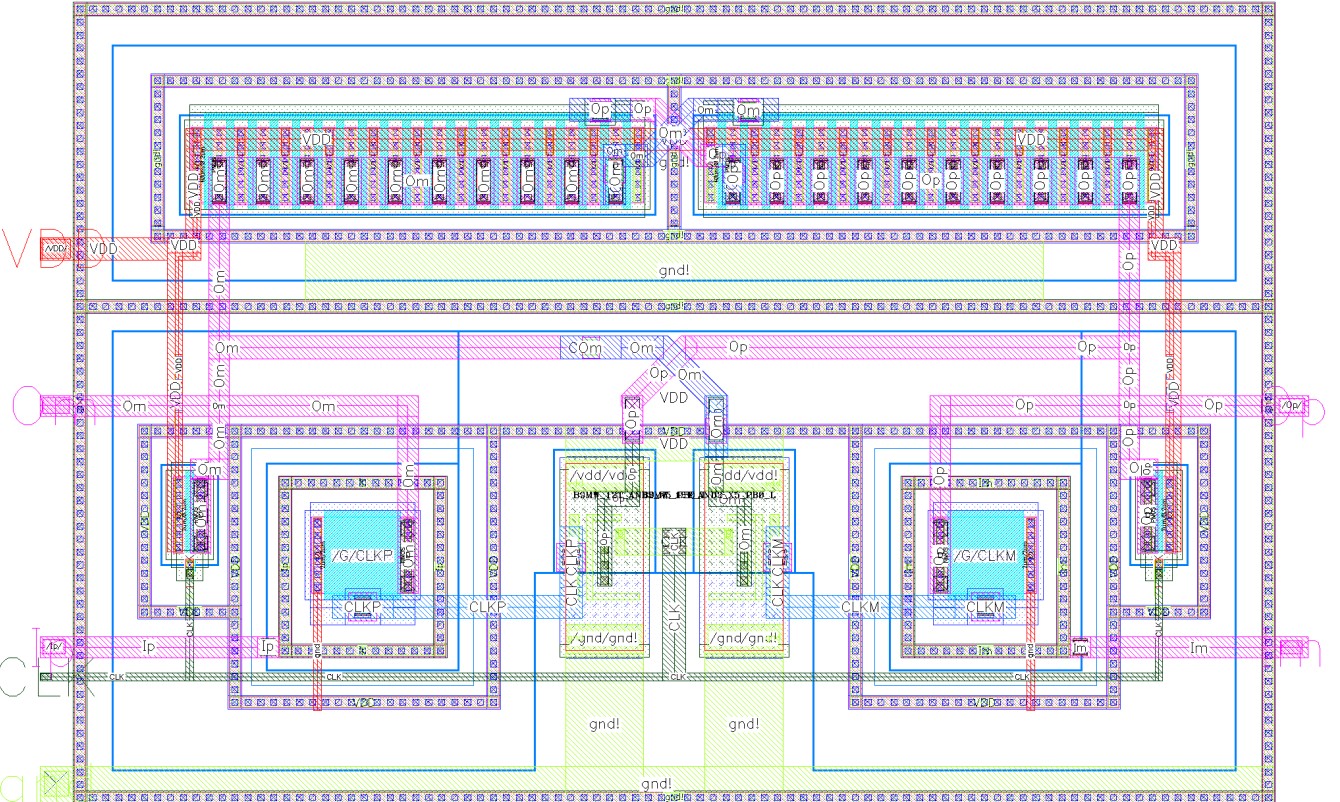

**Figure 5.** Layout of the proposed comparator.

Figure 6 shows the transient behavior of $V_p$, $V_q$, $V_{g1}$ and $V_{g2}$ during the evaluation phase. In accordance with the theoretical analysis in Section 3.1, during preamplification, the output nodes are discharged linearly because $I_{d1}$ and $I_{d2}$ remain constant. The initial spike that causes $V_p$ and $V_q$ to rise above $V_{DD}$ is due to clock feedthrough. Specifically, the clock edge is coupled to the outputs through the $C_{gd}$ of $M_{5,6}$ and through the equivalent parasitic capacitance between the AND gates' inputs. When the output common mode voltage reaches the threshold level, the positive feedback loop starts to regenerate the signal until the outputs saturate. For the whole duration of the preamplification phase and part of the regeneration phase, both $V_{g1}$ and $V_{g2}$ remain high. When $V_p$ becomes low enough, $V_{g2}$ toggles and turns off $M_2$, cutting the static consumption path formed by $M_2$ and $M_4$. Conversely, $V_{g1}$ remains high, thus, allowing for $V_q$ to be completely pulled down to ground.

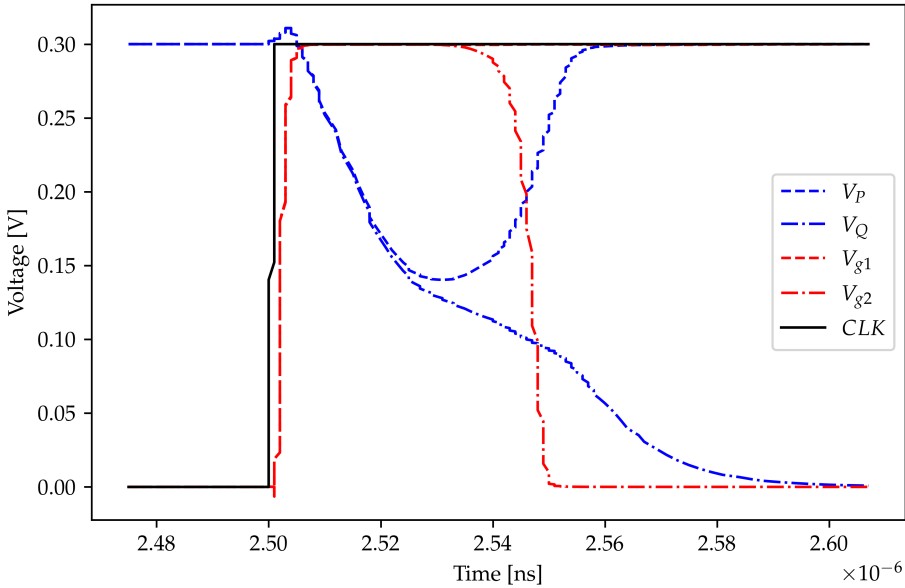

**Figure 6.** Transient behavior of $V_p$, $V_q$, $V_{g1}$ and $V_{g2}$ during the evaluation phase.

Figure 7 shows how the performance of the proposed comparator varies when the input common-mode level $V_{icm}$ is swept across the whole $[0, V_{DD}]$ range. The comparator exhibits rail-to-rail ICMR, as no heavy degradation in performance occurs at the extremities of the common mode range. Power consumption is almost unaffected, with a variation of less than 4%, while the comparison time experiences a variation of about 50%. The input-referred offset's standard deviation ($\sigma_{offset}$) also exhibits a total variation of about 50%, worsening as the input common-mode voltage increases. This behavior can be easily explained by considering that increasing the input common-mode voltage in an NMOS body-driven differential pair results in a lower threshold voltage, hence a larger overdrive (gate–source voltage is set to $V_{DD}$, that is larger than the nominal threshold voltage at 0.3 V supply), and the standard deviation of the input-referred offset is proportional to the overdrive voltage.

Table 2 summarizes the performance of the proposed comparator at $V_{id} = 1$ mV for three different choices of $V_{DD}$: 0.15 V, 0.3 V and 0.5 V. The maximum clock frequency able to guarantee operation in the worst-case corner was considered. Comparator delay and power dissipation are reported, as well as the power-delay product (PDP) and the energy-delay product (EDP). Thanks to the energy-efficient design that minimizes the number of stacked devices, the proposed topology achieves excellent performance in terms of both delay and power consumption. An inspection of the comparison speed reveals that even at $V_{DD} = 0.3$ V, the comparator can easily achieve clock frequencies above 1 MHz, and slower operation down to 0.15 V is guaranteed.

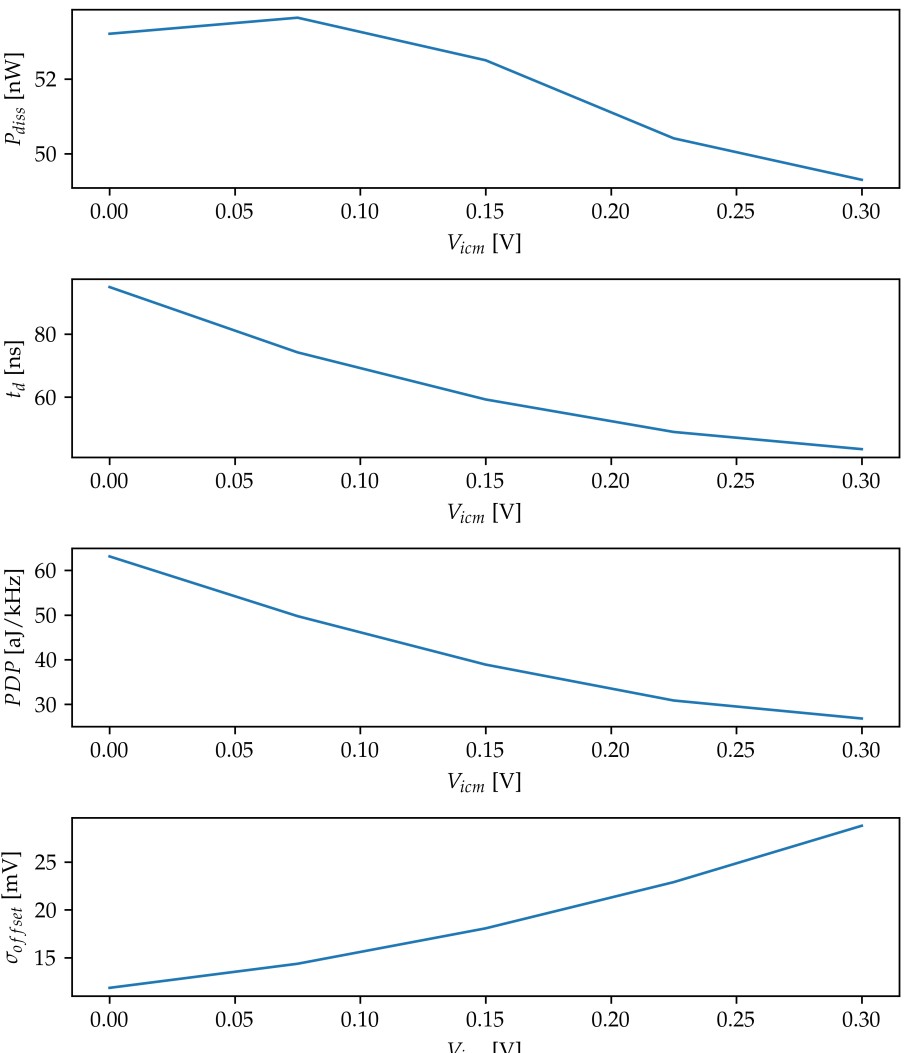

**Figure 7.** Performance of the proposed comparator when $V_{icm}$ varies from 0 to $V_{DD}$.

**Table 2.** Performance summary of the proposed comparator vs. $V_{DD}$, with $V_{id}$ = 1 mV.

| $\mathbf{V}_{DD}$ [V] | 0.15 | 0.3 | 0.5 |
|---|---|---|---|
| $f_{ck}$ [MHz] | 0.08 | 2 | 16 |
| $t_d$ [ns] | 2200 | 59.27 | 9.26 |
| $P_d$ [nW] | 0.856 | 52.50 | 12.96 |
| PDP [fJ] | 1.88 | 3.11 | 12.96 |
| EDP [aJ/kHz] | 23.584 | 1.556 | 0.810 |

To better highlight the ULV capabilities of the proposed comparator, Figure 8 shows how the comparator delay scales with the supply voltage. Reducing the supply voltage below 0.15 V results in excessive delays, whereas the upper limit to the supply voltage is set by the need to avoid turning on the body-source junction.

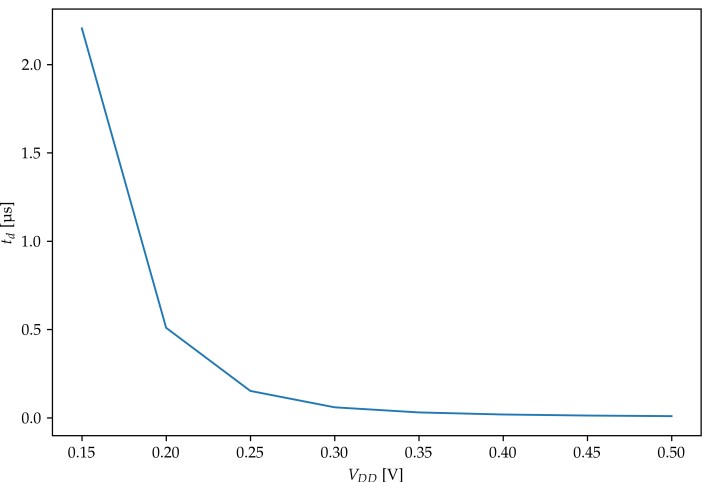

**Figure 8.** Delay of the proposed comparator vs. $V_{DD}$.

Since the body terminals can absorb significant amounts of current depending on the voltage that is applied to them, the static input currents were evaluated with DC sweep simulations to demonstrate that they remain limited during operation. To this end, the clock signal was set to $V_{DD}$, and the output voltages $V_P$ and $V_Q$ were set to the same value to deactivate the positive feedback loop. This was conducted to prevent the comparator from tripping during the DC simulation, which would have created an asymmetry in the final result. The output common mode was chosen so as to meet the condition $I_{d3,4} \approx I_{d1,2}$. Figure 9a,b shows the static input currents versus the input differential voltage when $V_{DD} = 0.3$ V and when $V_{DD} = 0.5$ V, respectively. At 0.3 V supply voltage, the input currents are negligible, as they do not even exceed 1 pA. Predictably, their value increases significantly when the supply voltage is scaled up to 0.5 V. The maximum current, in this case, is around 100 pA, which, however, is still acceptable for most applications. From Figure 9a, it can be seen that the sign of the current toggles when the single-ended input voltage falls below a certain threshold (which is around 100 mV). This is due to the fact that the body current is given by the difference between the forward current that flows in the source–body junction and the reverse current that flows in the drain–body junction. When $V_{BS}$ becomes small enough, the first term becomes smaller in magnitude than the second one. This, in turn, causes the current absorbed by the transistor's body terminal to become negative. For the sake of brevity, we avoided including plots of the static input currents versus $V_{icm}$. Indeed, the results are almost identical to those shown in Figure 9a,b, with the only difference that one of the two curves is reflected with respect to the *y*-axis.

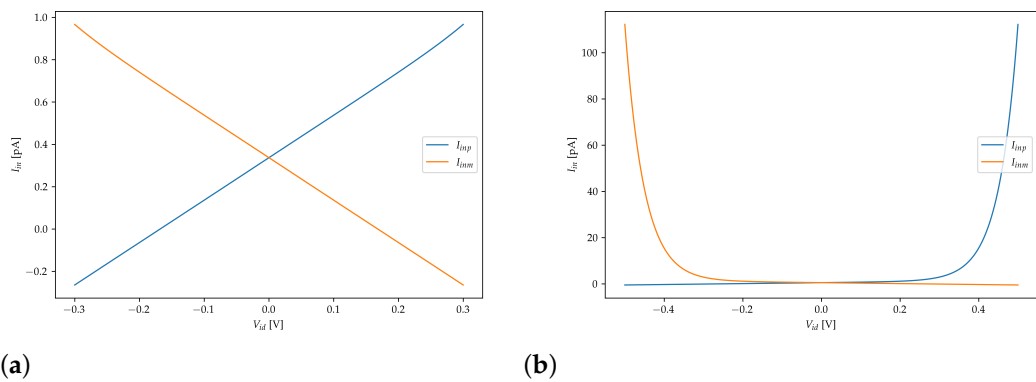

**(a)**　　　　　　　　　　　　　　　　　　　　**(b)**

**Figure 9.** Static input currents of the proposed comparator vs. input differential voltage with $V_{icm} = V_{DD}/2$ for **(a)** $V_{DD} = 0.3$ V; **(b)** $V_{DD} = 0.5$ V.

To assess the robustness of the proposed topology, extensive PVT (process, supply voltage and temperature) and mismatch analyses were performed. Table 3 summarizes the performance of the comparator under PVT variations at $V_{DD}$ = 0.3 V and $f_{ck}$ = 2 MHz, highlighting good performance consistency under a wide range of operating conditions. Power consumption remains quite constant since it is dominated by the dynamic component due to switching, whereas some variation of the delay is reported.

**Table 3.** Corners at $V_{DD}$ = 0.3 V, $f_{ck}$ = 2 MHz and $V_{id}$ = 1 mV.

| Performance | Nominal | Vddmin | Vddmax | Tmin | Tmax | SS | FF | SF | FS |
|---|---|---|---|---|---|---|---|---|---|
| $t_d$ [ns] | 59.27 | 97.94 | 39.4 | 74.09 | 44.85 | 81.78 | 44.38 | 79.97 | 49.08 |
| $P_d$ [nW] | 52.5 | 42.35 | 65.62 | 41.84 | 71.6 | 48.9 | 57.72 | 51.63 | 53.71 |
| PDP [fJ] | 3.11 | 4.15 | 2.59 | 3.10 | 3.21 | 4.00 | 2.56 | 4.13 | 2.64 |
| EDP [aJ/kHz] | 1.56 | 2.07 | 1.29 | 1.55 | 1.61 | 2.00 | 1.28 | 2.06 | 1.32 |

Figure 10a,b shows the results of the Monte Carlo simulations that were carried out to evaluate the comparator's power consumption and delay (respectively) under process variations. The results confirm the robustness of the circuit against process variations.

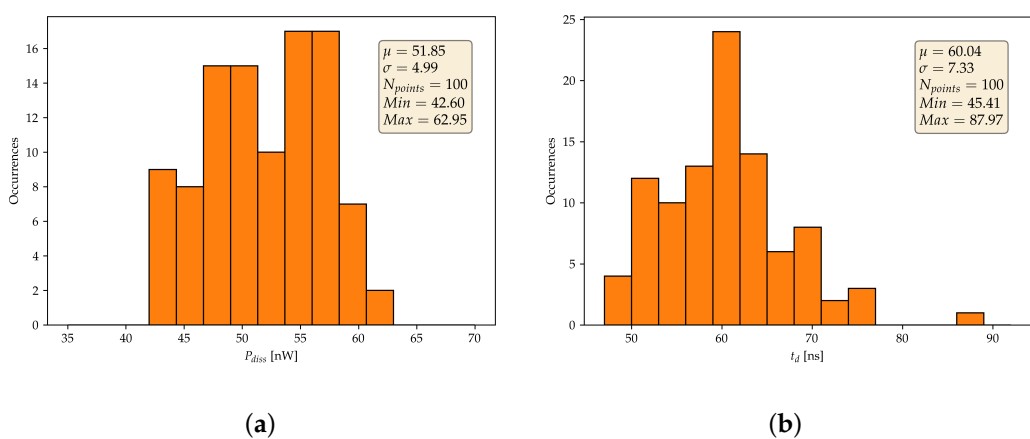

(**a**)         (**b**)

**Figure 10.** Histogram of (**a**) power consumption and (**b**) delay of the proposed comparator under process variations.

Figure 11 shows the histogram of the comparator's input-referred offset evaluated under 100 iterations of Monte Carlo mismatch simulation. The resulting standard deviation is about 18 mV, while the mean value is very small (< 1 mV). The standard deviation is higher with respect to gate-driven topologies because the body transconductance $g_{mb}$ is several times lower with respect to the gate transconductance $g_m$. This, however, is a common issue of body-driven comparators.

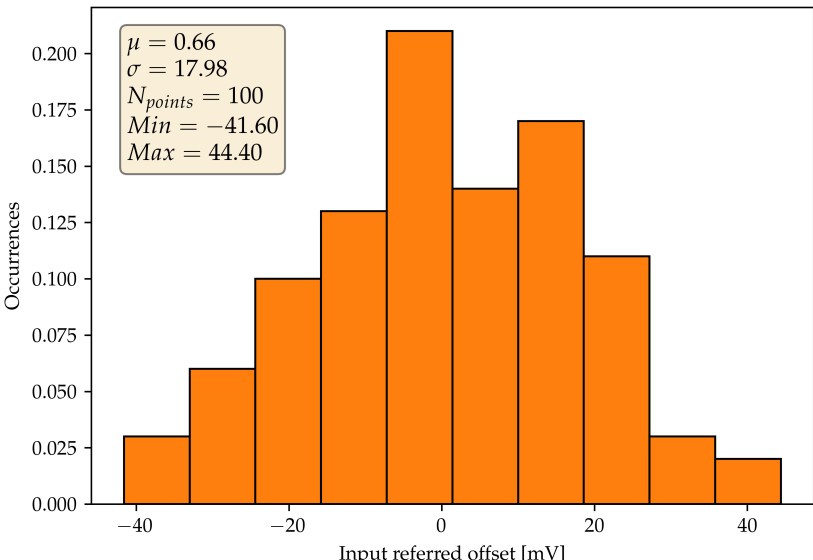

**Figure 11.** Histogram of input-referred offset under mismatch variations.

### 4.2. Application: Design of a SAR ADC

As a further validation, the proposed topology was simulated in a fully differential SAR ADC, whose schematic is shown in Figure 12. A 7-bit resolution was chosen because low-power sensors often rely on converters with medium-low resolution (6–8 bit) [18,64,65]. At the beginning of each conversion cycle, the input signal is sampled on a capacitive digital-to-analog converter (CDAC). The comparator then resolves the most significant bit (MSB), and the result is used to update the CDAC's output. This procedure is repeated until all the bits have been resolved. At the last decision, the digital code is sampled on the output register and is kept memorized while the ADC acquires and converts a new sample. The subsequent clock cycle is used to reset the control logic and the SAR registers. The control logic, the SAR registers and the hold register were implemented with standard library cells from the same 130-nm technology used for the comparator. The analog blocks (i.e., the sample-and-hold switches, the CDAC switches and the capacitors) were realized with ideal components to highlight better the impact of the comparator on the system's performance. A monotonic switching (also known as set-and-down) scheme was chosen for the CDAC [37]. Compared to the trial-and-error approach, monotonic switching halves the total capacitance of the CDAC, which saves power and area and improves the DAC's settling time. Moreover, no combinatorial control logic is needed, as the outputs of the SAR registers can directly drive the CDAC switches. The resulting topology is more efficient and easier to design compared to other switching schemes. The main drawback of the monotonic switching SAR algorithm is that the input common mode of the comparator converges to either $V_{DD}$ or ground (depending on how the CDAC is designed) during the conversion, which means that conventional comparator topologies cannot be used without compromising performances. In our case, however, monotonic switching can be used without negative consequences because the new comparator has rail-to-rail ICMR. Additionally, the CDAC can be specifically designed to have the comparator's input common mode converge to ground, which means that only NMOS switches operate during the conversion phase. This results in a highly optimized design because, for a fixed requirement on settling time, NMOS transistors have better area and power efficiency than PMOS transistors.

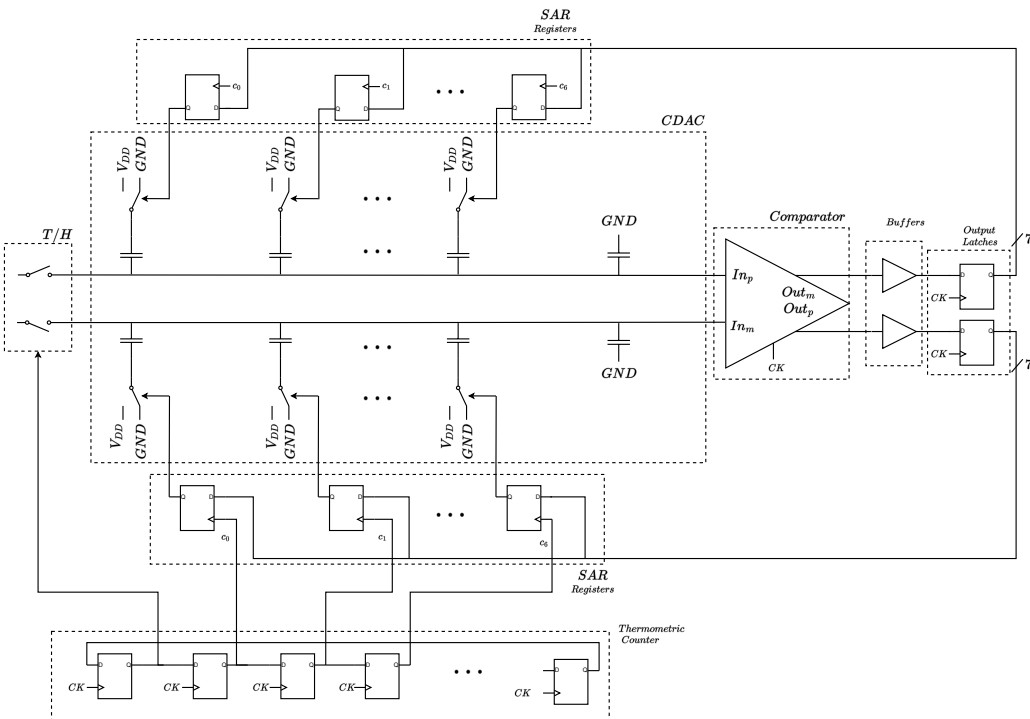

**Figure 12.** Schematic of the SAR ADC used to validate the proposed comparator.

The unit capacitance of the CDAC was set to 20 fF. The converter operates at a clock frequency of 1 MHz, which corresponds to a sampling frequency $f_s \approx 111$ kS/s, and its peak-to-peak differential input swing is 580 mV. The ADC was tested by running a transient noise simulation with a sinusoidal input at full-scale amplitude and at a frequency $f_{sine} = f_s/32$. Figure 13 shows one period of the sine waveform converted by the ADC. Figure 14 shows the same signal in the frequency domain. The spurious free dynamic range (SFDR) is 44.22 dB. The signal-to-noise-and-distortion ratio (SNDR) is 39.24 dB, which corresponds to an effective number of bits (ENOB) of 6.24 bits. A significant part of the loss in accuracy is caused by the comparator's noise. Indeed, without noise, the simulated SNDR is 41.96 dB, which corresponds to an ENOB of 6.68 bits.

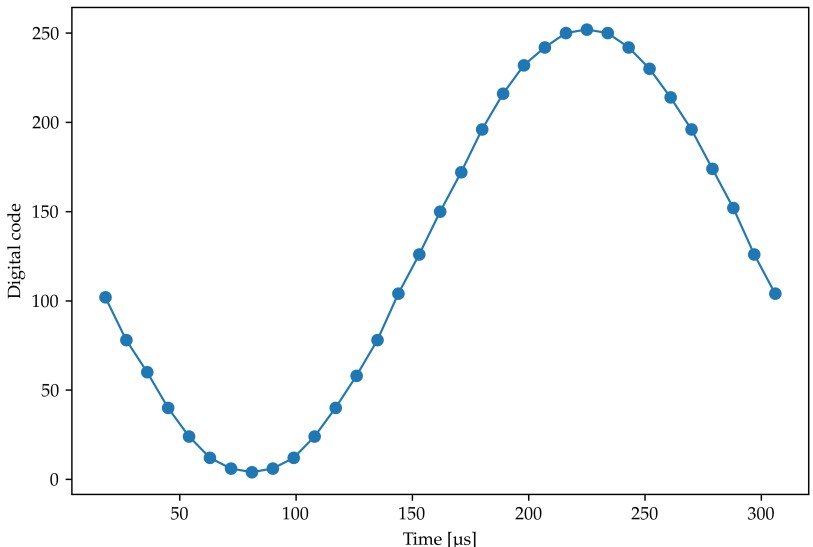

**Figure 13.** Time domain output of the ADC.

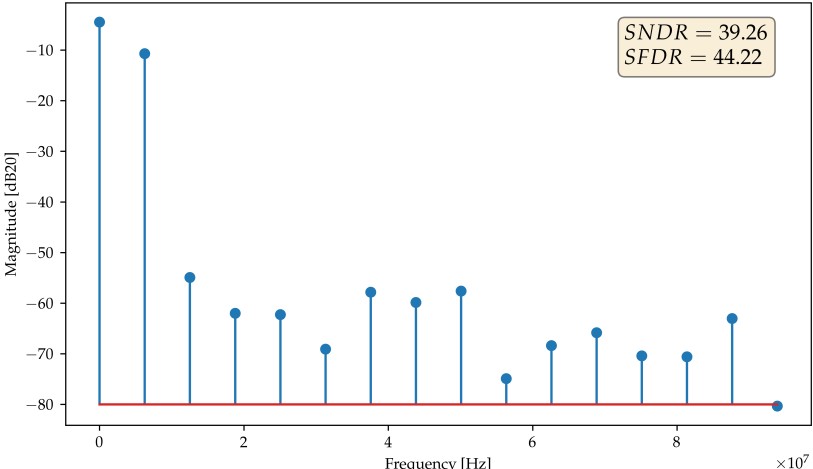

**Figure 14.** Spectrum of the ADC output.

## 5. Comparison

This section presents a comparison of the performance of the proposed topology with the state-of-the-art of ultra-low voltage comparators. It has to be noted that even if a number of comparators operating at 0.5 V or lower have been reported in applications, very few of them have been characterized.

To allow a comparison among different designs, suitable figures of merit (FOMs) have to be used. In particular, a possible FOM for comparators is the energy-delay product (EDP), which evaluates how fast a comparator is for a given power consumption and clock frequency and is defined as:

$$EDP = \frac{P_d \, t_d}{f_{ck}} \tag{19}$$

Clearly, the lower EDP, the better the trade-off between power consumption and delay.

In addition, since the input common mode range is an important parameter in some applications, we propose a further FOM defined as the EDP divided by the ICMR normalized to the supply voltage:

$$FOM = \frac{P_d}{ICMR/V_{DD}} \cdot \frac{t_d}{f_{ck}} = \frac{EDP \cdot V_{DD}}{ICMR} \tag{20}$$

Moreover, in this case, a lower FOM indicates a better performance taking into account both the speed-power trade-off and the input common-mode range. Because the normalized ICMR appears at the denominator, the proposed FOM favors comparators with a larger ICMR (relative to the supply voltage). Normalizing the ICMR ensures a fair comparison between topologies that have been tested at different supply voltages.

Table 4 reports the performance of comparators operating at 0.5 V or lower supply voltages; both analog comparators and standard-cell-based ones have been taken into account since the latter are often used in applications at these low supply voltages. Performance of the proposed comparator was reported at 0.15, 0.3 and 0.5 V supply, and similarly also, for the comparators in the literature, data were reported at different supply voltages if available. Table 4 reports for each comparator the main performance parameters (maximum delay $t_d$, power consumption $P_d$, offset) together with the simulation conditions (input differential voltage $V_{id}$ and clock frequency $f_{ck}$). The table also includes the type of input interface (gate-driven (GD) or body-driven (BD)) and the comparator topology. Further performance parameters are area, supply voltage and ICMR. This allows calculating the power-delay product (PDP), the EDP and the FOM defined in (20).

**Table 4.** Comparison with the state-of-the-art of ultra-low voltage comparators.

| | This Work | | | [58] | [58] | [48] | [61] | [46] | [41] | [66] | [48] | [45] | [46] |
|---|---|---|---|---|---|---|---|---|---|---|---|---|---|
| Topology | BD-SA | | | STD-CELL | STD-CELL | BD-SA | STD-CELL | BD-SA | DT | DT | BD-SA | SA | BD-SA |
| $V_{DD}$ [V] | 0.15 | 0.3 | 0.5 | 0.15 | 0.3 | 0.3 | 0.35 | 0.35 | 0.4 | 0.4 | 0.5 | 0.5 | 0.5 |
| Technology [nm] | 130 | 130 | 130 | 180 | 180 | 180 | 45 | 90 | 180 | 28 | 180 | 180 | 90 |
| Type | BD | | | GD | GD | BD | GD | BD | BD | GD | BD | GD | BD |
| Area [μm$^2$] | 670 | 670 | 670 | 900 | 900 | - | 59 | - | - | - | - | - | - |
| ICMR [mV] | 150 | 300 | 500 | 135 | 275 | 300 | 350 | - | 200 | 200 | 500 | - | 500 |
| ICMR-rail-to-rail | ✓ | ✓ | ✓ | ✓ | ✓ | ✓ | ✓ | ✓ | ✗ | ✗ | ✓ | ✗ | ✓ |
| Max. $t_d$ [ns] | 2200 | 59.27 | 9.26 | 442,000 | 34,700 | 980 | 2100 | 2.22 | 594 | 1330 | 16.4 | 5.77 | 0.5 |
| $V_{id}$ [mV] | 1 | 1 | 1 | 10 | 10 | 0.5 | 10 | - | 0.1 | 0.1 | 0.5 | 2 | 1 |
| $f_{ck}$ [kHz] | 80 | 2000 | 16,000 | 10 | 10 | 62.5 | 10 | 50 | 100 | 100 | 5000 | 200,000 | 333,000 |
| Offset ($\sigma_{off}$) [mV] | 18.74 | 17.98 | 18.14 | 31 | 8 | - | 4.73 | - | 13.7 | 15.3 | - | 0.29 | 5.1 |
| $\frac{\sigma_{off}}{V_{DD}}$ [%] | 12.49 | 5.99 | 3.63 | 20.67 | 2.67 | - | 1.35 | - | 3.42 | 3.82 | - | 0.06 | 1.02 |
| $P_d$ [nW] | 0.856 | 52.5 | 1400 | 0.027 | 0.024 | 0.1 | - | 184 | 4.48 | 14.6 | 20.2 | 34,000 | 2300 |
| PDP [aJ] | 1883.86 | 3111.67 | 12,964 | 11,934 | 3088.3 | 98 | - | 408.48 | 2661.12 | 19418 | 331.28 | 196,180 | 1145.4 |
| EDP [aJ/kHz] | 23.55 | 1.55 | 0.81 | 1193.4 | 308.83 | 1.57 | - | 8.17 | 26.61 | 194.18 | 0.07 | 0.98 | 0.003 |
| FOM [pJ/kHz] | 23.55 | 1.55 | 0.81 | 1326 | 336.9 | 1.57 | - | - | 53.22 | 388.36 | 0.07 | - | 0.003 |

SA: StrongARM; BD: Body-Driven; GD: Gate-Driven; BD-SA: Body-Driven StrongARM; DT: Double-Tail; STD-CELL: Standard-Cell-Based.

The only other reported comparator operating at 0.15 V has been presented by [58] (comparators in [16] and [47] were not characterized) and presents a much higher delay, resulting in worse figures of merit. Offset is also larger, notwithstanding gate driving and a larger area. The comparison table also shows that the proposed comparator outperforms the other comparators in the literature up to 0.4 V supply voltage. A similar EDP at 0.3 V was reported by [48], which, however, reports no data on offset performance. Both the comparator in [48] and [46] are efficient at the 0.5 V supply.

## 6. Conclusions

In this paper, we propose an ultra-low voltage latched comparator based on the StrongARM topology. Body driving is exploited to achieve rail-to-rail ICMR; inverters in the standard StrongARM topology are substituted by simple cross-coupled PMOS devices, and the clocked tail current generator is eliminated, achieving a core comparator with just two stacked devices. The clock signal is applied to the gates of the input devices to turn them on and off through a feedback path that allows for minimizing power consumption.

The comparator can operate at supply voltages as low as 0.15 V with excellent performance in terms of speed and power consumption and an input-referred offset of about 18 mVrms comparable with other body-driven topologies. A comparison with the literature highlights that the proposed comparator outperforms the state-of-the-art for supply voltages up to 0.4 V, showing at 0.15 V (0.3 V) a power consumption of 856 pW (52.5 nW) with an operating clock frequency of 80 kHz (2 MHz) and a maximum delay of 2.2 µs (59.27 ns).

**Author Contributions:** Conceptualization, R.D.S. and V.S.; methodology, R.D.S., V.S. and F.C.; software, R.D.S., V.S. and C.B.; validation, R.D.S., V.S., and F.C.; formal analysis, R.D.S. and V.S.; investigation, R.D.S., V.S. and C.B.; resources, F.C. and A.T.; data curation, R.D.S., V.S. and C.B.; writing—original draft preparation, R.D.S., V.S. and C.B.; writing—review and editing, F.C.; visualization, A.T.; supervision, F.C. and A.T.; project administration, F.C. and A.T. All authors have read and agreed to the published version of the manuscript.

**Funding:** This research received no external funding.

**Data Availability Statement:** The data presented in this study are available in article.

**Conflicts of Interest:** The authors declare no conflict of interest.

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
