# Peer review of "A 0.15-to-0.5 V Body-Driven Dynamic Comparator with Rail-to-Rail ICMR"

_jlpea, doi:10.3390/jlpea13020035_

Round 1

Reviewer 1 Report

The paper presents a novel dynamic body-driven ultra-low voltage comparator. The introduced topology allows for the removal of the clocked tail generator, where the number of stacked transistors decreases and performance at low voltage is improved. The introduction part is well-written with a detailed literature review. Details of the comparator are presented in Section 2. The comparator delay is analyzed in section 3, and Section 4 presents simulation results. 

Please make sure that Equation (6) is correct. (g_ma)

How have the transistor sizes been chosen as Table 1? It should be described.

In Figure 6, Why do the V_P and V_Q exceed the CLK signal between 2.50 and 2.503 ns?

Please double-check lines 232-235.

Table 4 should be explained in more detail in Section 5. 

The authors proposed a further FOM defined as the EDP divided by the ICMR normalized to the supply voltage. This term should be presented more. 

Author Response

The paper presents a novel dynamic body-driven ultra-low voltage comparator. The introduced topology allows for the removal of the clocked tail generator, where the number of stacked transistors decreases and performance at low voltage is improved. The introduction part is well-written with a detailed literature review. Details of the comparator are presented in Section 2. The comparator delay is analyzed in section 3, and Section 4 presents simulation results.

  • Please make sure that Equation (6) is correct. (g_ma)

REPLY: We thank the reviewer for this comment. We reviewed all the subsection 3.2 on the analysis of the comparator in the regeneration phase and modified Fig. 4 accordingly. Now eq. (6) – eq. (9) in the revised manuscript – should result coherent with Fig. 4 and with the description of the analysis (it is not a small-signal analysis, but a transient analysis in the Laplace domain with initial conditions using linearized models).

  • How have the transistor sizes been chosen as Table 1? It should be described.

REPLY: We sized the transistors taking into account the trade-off between speed and performance, and with the goal of balancing the transconductance of NMOS and PMOS devices. We have added a discussion on this issue to the manuscript.

  • In Figure 6, Why do the V_P and V_Q exceed the CLK signal between 2.50 and 2.503 ns?

REPLY: Figure 6 shows the transient behavior of several voltage waveforms. In particular, voltages at nodes P and Q present some overshoot and exceed the CLK signal due to the effect of clock feedthrough. This point has been better clarified in the revised manuscript.

  • Please double-check lines 232-235.

REPLY: We thank the reviewer for this comment. In the reported simulation, Vgs=VDD=0.3V is larger than the nominal threshold voltage; increasing the input common mode voltage (i.e. the Vbs voltage) lowers the threshold voltage and increases the overdrive of the NMOS devices. This results in a larger standard deviation of the input-referred offset, as is well known by the theory. The situation is more complicated for lower supply voltages, when the devices are biased in weak inversion. In the revised manuscript, we have modified the sentence trying to make it clearer.

  • Table 4 should be explained in more detail in Section 5.

REPLY: We thank the reviewer for this comment. In the revised manuscript we have commented more in detail the comparison table (Tab. 4) to better highlight the advantages of the proposed topology.

  • The authors proposed a further FOM defined as the EDP divided by the ICMR normalized to the supply voltage. This term should be presented more.

REPLY: A large ICMR is important in some comparator applications, in particular in a ultra-low voltage environment. To take this into account, we have defined a figure of merit (FOM) that also includes the ICMR (normalized to the supply voltage) together with the EDP (energy delay product), that measures the power efficiency. We have better clarified this point in the revised manuscript.

Reviewer 2 Report

The authors of this work have proposed a bulk-driven comparator. To improve the quality of the manuscript, the following concerns should be clarified.

1-      The delay of the proposed bulk-driven comparator should be compared with a similar gate-driven one.

2-      The target application for the proposed comparator especially under ultralow-voltage supplies should be described and the circuit should be employed in an application to show its efficiency.

3-      Since a comparator usually receives a large input signal, the input current though bulk terminal should be evaluated to see if the input impedance and the operation are affected or no.

4-      The Monte-Carlo analysis should also be done for other parameters such as delay, decision-making capability, etc. to see if the operation of the comparator is sensitive to the mismatch or no.

5-      It should be mentioned and discussed that the area of the proposed comparator has increased due to bulk terminal isolations and …

6-      Since the transistors turn on in the saturation or triode in the weak inversion region, why a gate-driven structure can not be better than bulk-driven one? Why the input common-mode voltage in a comparator should be important while a gate driven structure is also able to decides in the weak-inversion region. Please clarify through the manuscript.

7-      Regarding the input offset, please add the following reference to the manuscript and a verbal explanation of the input offset of the proposed comparator should be added according to: "Input Offset Estimation of CMOS Integrated Circuits in Weak Inversion," in IEEE Transactions on Very Large Scale Integration (VLSI) Systems, vol. 26, no. 9, pp. 1812-1816, Sept. 2018.

Author Response

The authors of this work have proposed a bulk-driven comparator. To improve the quality of the manuscript, the following concerns should be clarified:

  • The delay of the proposed bulk-driven comparator should be compared with a similar gate-driven one:

REPLY: In the manuscript, we compared the proposed bulk-driven comparator with those from literature, including those exploiting a gate-driven input. In order to make this point clearer, we have added to the revised manuscript a further row in the comparison table in which we specify if the architecture is gate-driven or body-driven.

 The target application for the proposed comparator especially under ultralow-voltage supplies should be described and the circuit should be employed in an application to show its efficiency:

REPLY: We thank the reviewer for this observation. As discussed in the Introduction, latched comparators in a ultra-low voltage environment find application for example in ADCs and LDOs. In the revised manuscript we have added as a case study a fully differential SAR ADC. The set-and-down algorithm was adopted, since it provides a net reduction of area and power consumption, but presents a common mode voltage of the comparator that is not constant along the SAR iterations.

  • Since a comparator usually receives a large input signal, the input current though bulk terminal should be evaluated to see if the input impedance and the operation are affected or no.

REPLY: We agree with the reviewer that body driving could be an issue on some applications; on the other hand, we operate in an ultra-low voltage environment, hence forward biasing of the bulk-channel junction is limited. In the revised manuscript we added Figure 9 to show the bulk current vs input voltage; it shows that current remains below 1pA at 0.3V supply, and below 100pA at 0.5V supply. We further discuss the bulk current issue in the revised manuscript.

  • The Monte-Carlo analysis should also be done for other parameters such as delay, decision-making capability, etc. to see if the operation of the comparator is sensitive to the mismatch or no:

REPLY: We thank the reviewer for his observation. We have expanded the simulation section by adding Monte Carlo simulations for process variations. We also report in the attached document the histograms of power consumption and delay under mismatch variations. However, we chose not to include these simulations in the manuscript, because we believe they provide limited insight on the circuit's performance: indeed, a comparator affected by mismatch is equivalent to an ideal comparator with an input-referred offset V_offset superimposed to its input differential voltage. In a Monte Carlo simulation, each iteration returns the delay and power consumption evaluated at an effective input difference V_id,eff = V_id + V_offset (V_offset being a random variable), rather than at a known V_id. This causes the delay distribution to become narrower when the standard deviation of the offset is large enough compared to V_id, because in this case |V_id,eff| will be, on average, larger than |V_id|. The same holds true for the distribution of power consumption, because the comparator dissipates slightly less energy on "easier" decisions. This also explains why, in the simulations, the mean values of delay and power consumption are smaller compared to the values obtained in the nominal corner.

  • It should be mentioned and discussed that the area of the proposed comparator has increased due to bulk terminal isolations and …

REPLY: We thank the reviewer for this comment. Surely the need to isolate the bulk terminals of NMOS devices requires additional area. However this is already considered in the layout of figure 5. We have added a comment on this issue in the revised manuscript.

  • Since the transistors turn on in the saturation or triode in the weak inversion region, why a gate-driven structure can not be better than bulk-driven one? Why the input common-mode voltage in a comparator should be important while a gate driven structure is also able to decides in the weak-inversion region. Please clarify through the manuscript.

REPLY: While we agree with the reviewer that a gate-driven comparator working in weak inversion could function at 0.3V supply, we would like to emphasize that many applications require a more or less large ICMR, that could not be compatible with gate driving, unless some complementary approach is adopted, as in the standard-cell-based topologies. Furthermore, in a ULV environment, “small” variations of the input common mode voltage are a large fraction of the supply voltage, thus complicating the situation. We have tried to clarify this issue in the Introduction, and we have added as an example application a SAR ADC exploiting the set-and-down algorithm, that requires the ICMR to cover half of the supply voltage range.

  • Regarding the input offset, please add the following reference to the manuscript and a verbal explanation of the input offset of the proposed comparator should be added according to: "Input Offset Estimation of CMOS Integrated Circuits in Weak Inversion," in IEEE Transactions on Very Large Scale Integration (VLSI) Systems, vol. 26, no. 9, pp. 1812-1816, Sept. 2018.

REPLY: We added this reference in the revised manuscript and we derived a relation between gate-driven and body-driven input referred offset according to the suggested reference.

Reviewer 3 Report

In this paper, authors presented a novel dynamic body-driven ultra-low voltage (ULV) comparator. They have prepared a well-discussed work which presents a well-considered up-to-date aspect which has interest for all. The paper is good writing, and there is no necessary anyone modification. In my opinion, it could be accepted for publication.

Moderate editing of English language.

Author Response

In this paper, authors presented a novel dynamic body-driven ultra-low voltage (ULV) comparator. They have prepared a well-discussed work which presents a well-considered up-to-date aspect which has interest for all. The paper is good writing, and there is no necessary anyone modification. In my opinion, it could be accepted for publication.

REPLY: We are very grateful to the reviewer for this kind comment.

Round 2

Reviewer 2 Report

All my concerns have been addressed.